# Traumatic Aneurysm Involving the Posterior Communicating Artery

**DOI:** 10.3390/healthcare12020192

**Published:** 2024-01-13

**Authors:** Gabriele Napoletano, Nicola Di Fazio, Giuseppe Delogu, Fabio Del Duca, Aniello Maiese

**Affiliations:** 1Department of Anatomical, Histological, Forensic and Orthopedic Sciences, Sapienza University of Rome, 00161 Rome, Italy; nicola.difazio@uniroma1.it (N.D.F.); giuseppe.delogu@uniroma1.it (G.D.); fabio.delduca@uniroma1.it (F.D.D.); 2Department of Surgical, Medical and Molecular Pathology and Critical Care Medicine, Section of Legal Medicine, University of Pisa, 56126 Pisa, Italy; aniello.maiese@unipi.it

**Keywords:** traumatic brain injury, forensic neuropathology, traumatic intracranial aneurysm, cerebral angiography, aneurysm rupture

## Abstract

Traumatic intracranial aneurysms (TICAs) are rare, accounting for less than 1% of all intracranial aneurysms. However, they are associated with a mortality rate of over 50%. The case presented herein focuses on a posterior communicating artery TICA caused by violent aggression. A 41-year-old man with massive subarachnoid hemorrhage (SAH), on admission to hospital, had a CT angiography that showed a ruptured left posterior communicating artery aneurysm with continuous blood loss and underwent neurosurgical cooling. The CT scan also showed fractures of the mandible, mastoid and left styloid process, as well as brain contusions caused by blows and kicks. Despite medical treatment and surgery, after four days, he died. The assault dynamics were recorded by a camera in the bar. The damage was caused by kicks to the neck and head. The forensic neuropathological examination showed the primary injury (SAH, subdural hemorrhage, cerebral contusions, head–neck fractures), as well as secondary damage following the attack (cerebral infarcts, edema, supratentorial hernia, midbrain hemorrhage). The coil was intact and well positioned. In this case, circumstantial information, medical records, and the type of injury could shed light on the mechanism of the production of a TICA. In addition, the CT angiography and histological investigations helped to distinguish a recent and traumatic aneurysm from a pre-existing one. Following precise steps, the study of aneurysms can be helpful in clarifying their traumatic origin even when the victim was taking drugs. The aim of this study is also to share the diagnostic process that we used in the forensic field for the assessment of suspected traumatic aneurysms.

## 1. Introduction

Traumatic intracranial aneurysms (TICAs) are rare, accounting for less than 1% of all cerebral aneurysms and are associated with a mortality rate of more than 50% [1]. They may develop following blunt or penetrating trauma and a small percentage of them may involve the posterior communicating arteries. Histologically, TICAs can be classified as true aneurysms, pseudo-aneurysms (false aneurysms), mixed aneurysms, and dissecting aneurysms (hematoma) [2,3]. A true aneurysm is a local dilation resulting from focal wall weakness, and the aneurysm wall is formed by the same layers as the main vessel [3]. Pseudo-aneurysms are due to a complete tear in the vessel wall with the formation of a hematoma that undergoes fibrous organization and remains in communication with the vessel lumen. The latter occur often after trauma and may produce delayed subarachnoid hemorrhage (SAH). Mixed aneurysms form when the rupture of a true aneurysm causes the secondary formation of a pseudo-aneurysm. Dissecting hematomas, on the other hand, are due to a dislocation between the lamina intima and the media with the formation of a false lumen that can rupture with distension [3]. Blood from the false lumen can rupture internally, where there is no prospect of dilation, or externally, with the potential development of a false aneurysm. A percentage of 80% of ruptured aneurysms are found in the anterior circulation and only 20% in the posterior circulation [4,5]. In head injuries with subarachnoid or intraparenchymal hemorrhage, cerebral contusions, axonal injury due to acceleration/deceleration and fractures, the formation or rupture of an aneurysm can occur as a result of the stretching of cerebral vessels and pressure changes [4,6]. In such severe cases, it may be easier to argue a casual correlation between aggression and aneurysm rupture. However, the rupture of an aneurysm can also occur in the case of minor trauma, when there are no body injuries, but the trauma has resulted in the abrupt rotation of the head and neck [4,6]. In a medico-legal context, the assessment of such a case becomes more complex if the patient suffers from arterial hypertension, takes medication, or smokes tobacco or other drugs potentially capable of rupturing an aneurysm and increasing the risk of death [7]. SAH is often located in the basal brain and the aneurysm is often invisible. A careful neuropathological examination is essential to assess whether the cause of SAH is a traumatic aneurysm or a pre-existing aneurysm.

## 2. Case Report

A 41-year-old male assault victim was autopsied, after the detailed examination of clinical reports.

The man was punched and kicked in the head multiple times. The scene was witnessed and recorded. Then, he was taken to the hospital in a comatose state. In the emergency room, CT and CT-angiography (Figure 1) were performed; diffuse subarachnoid, subdural hemorrhage, and areas of contusion in association with fractures of the bilateral mandible, styloid bone, and left mastoid process were highlighted. On CT angiography, a traumatic aneurysm of the left posterior communicating artery, blister-like, was bleeding. At the time of admission, toxicological investigations were performed which were positive for cocaine [0.224 mg/L] and ethanol [1.35 g/L]. The cocaine was at a low dose. It was consumed some hours before the fight and the loss of consciousness occurred immediately after the kicks to the head. This alcohol concentration would be consistent with drunkenness, loss of consciousness and motor coordination as well as irascibility. Several left ribs were found to be fractured as well. The patient was treated pharmacologically because of the high surgical risk, and on the following day, coiling of the internal carotid artery and branches of the left posterior and anterior communicating arteries was inserted. On the next days, radiological examinations showed areas of contusion and edema of the left hemisphere (>frontal–basal, temporal) with parahippocampal and basal nuclei hemorrhages, axonal damage at the level of the corpus callosum, and the flattening of the left lateral ventricle. Furthermore, on the right frontal cortex, contrecoup injury were visible. On the third day, vasospasm and ischemic injury of the basal parieto-occipital cortex were revealed by magnetic resonance imaging (MRI) (DTI). On the fourth day, brain death was confirmed.

Under the guidance of radiological images, the skull was removed. The brain must be examined and characterized. Macroscopic examination was conducted evaluating the epidural, dural, and subdural tissue integrity, as well as the skull and galea aponeurotica.

In cases of confirmed premortem cerebral hemorrhage or suspected cerebral hemorrhage, organ evisceration during autopsy is performed prior to the opening of the cranial vault. After the skull was removed, the subarachnoid hemorrhage at the base of the brain was thoroughly rinsed.

The dissection of the brain begins with the initial dissection of the *dura mater*, starting from the *crista galli*. Blood was observed within the basal cisterns. At this point, the optic nerves, oculomotor nerves, and abducens nerves were bilaterally sectioned. After tentorium dissection, a semilunar incision on medulla oblongata allowed the dissector to eviscerate the brain.

The brain was suspended using the basilar artery and immersed in 10% neutral buffered formalin for 3 weeks to allow for neuropathological examination. 

After fixation, before dissecting the brain, the Circle of Willis was carefully separated. Once the location of the aneurysm was identified, serial samples of approximately 0.5 mm were taken from the vessels of greatest interest. Representative brain samples were taken. All tissue blocks were processed and embedded in paraffin. After microtome cutting, the slides were stained with hematoxylin–eosin (H&E), Prussian blue (Perl’s), and Masson’s trichrome. All sections were also immunostained with an antibody against CD68 for microglia/macrophages and CD15 for neutrophils. For the medico-legal assessment, the documentation of the admission with the relevant instrumental CT and angio-CT examinations were acquired.

The autopsy revealed an extensive subarachnoid hemorrhage, with a subdural portion, as well as a small, rounded subpial hemorrhage located above the left cortical contusion area. The brain was swollen, with flattened gyri and narrow sulci. On neuropathological examination of the fixed brain, the lesions were confirmed (Figure 2A). Coronal sections showed hemorrhages of the corpus callosum and thalamic nuclei (Figure 2B). The examination of the vessels of the Circle of Willis showed a large blood clot involving the left internal carotid artery and the posterior communicating artery, as well as the presence of a metal coil (Figure 2C). Macroscopically, the posterior communicating artery was ruptured and the internal carotid artery dissected. Within the midbrain, extensive reddish-black hemorrhagic areas were found, indicative of recent hemorrhages (Figure 1D). The edema and increased supratentorial pressure led to hemorrhages, also known as “Duret” hemorrhages, which led to death. Microscopically, the vessels showed both the presence of an aneurysm of the posterior communicating artery and extensive infiltration between the elastic lamina and the connective layer associated with inflammatory cells (Figure 3A). Neutrophils, red blood cells, and fibrin were within the point of the rupture of the aneurysm, but there was no hemosiderin (Figure 3B,C). Samples of the left temporal and parietal cortex showed areas of necrosis, hemorrhage, and edema with neutrophils inside (Figure 3D). In the basal nuclei (Figure 3E) and contralateral cortex, similar findings were observed. The brain changes following a violent attack were the direct consequence of a traumatic aneurysm. Therefore, the case was classified as a homicide.

## 3. Discussion

The rupture of a pre-existing aneurysm or a previously healthy vessel can be caused by aggression. In cases like this, macroscopic neuropathology examination is very important to ensure targeted microscopic findings [8,9,10], because it is difficult to assess whether the rupture of the aneurysm is due to a stress-related increase in blood pressure or by blunt trauma, or the consequence of a mixed mechanism [11]. Furthermore, assessing the cocaine and benzoylecgonine concentration can assist with the interpretation of intracranial hemorrhage in drug abusers [12]. 

In closed head trauma, TICA can form by compression or stretching at the site of blows or in brain areas near rigid intracranial structures, such as skull bones (e.g., sphenoid bone) or meninges (falx and tentorium) [3]. In the rupture of TICAs, it is very important that there is a chronological continuity (i.e., a clear relationship between the assault and the onset of symptoms), so that a causal correlation between the two events can be sustained [13]. In our case, on the one hand, the dynamics of the aggression were known. On the other hand, the injuries (bruises, intracranial hemorrhages, fractures concentrated in the left mandibular angle and left mastoid) were compatible with a violent rotational trauma of the neck and head. In addition, the immediate loss of consciousness occurred (GCS 3). As it is well known, even a slight rotational trauma could have led to the formation and rupture of aneurysms [3,11,14]. Performing CT angiography and Digital Subtraction Angiography (DSA) a few hours after the attack not only has a diagnostic–therapeutic role, but can be useful tool to carry out medico-legal assessment [15]. Histological investigations are a further tool to clarify the nature of the aneurysm [16]. In fatal cases, they can be useful to verify whether it was the rupture of a newly formed or pre-existing aneurysm [11,16]. Finally, in addition to the local inflammatory response, the presence of hemosiderin deposits—which are indicative of a previously established inflammatory process—should be checked. In our case, the Masson’s stain and H&E showed inflammatory cells at the site of the aneurysm rupture. Fibrin deposits, neutrophil cells, and some within the clot formed around the ruptured vessel were detected. Conversely, the absence of hemosiderin deposits (H&E and Perl’s) and the presence of well-stained elastic lamina (Masson’s trichrome), despite the subject’s use of cocaine and alcohol, could suggest a recent TICA. Furthermore, small lacerations of the vessel layers infiltrated by fresh blood cells were highlighted (Figure 2C). The formation of aneurysms is associated with smoking and drug abuse [17,18,19,20,21]. This is influenced by hormonal factors, genetic conditions, and neuroinflammatory processes [17,18,19,20,21]. Aneurysms often go undetected for years, allowing hemosiderin to settle inside the vessel wall. Aneurysm remodeling can last years and, in rare cases, even result in macroscopic structural alterations, such as in large or giant aneurysms [22,23]. Drugs such as cocaine and methamphetamine predispose patients to rapid aneurysm formation and rupture arising from a sudden increase in blood pressure [24,25,26,27]. However, in assault with either major or minimal injury, with neck injuries from rotational trauma, most authors are inclined to speak of TICA rupture [3,4,6,11,14], especially when chronological continuity is reported. The secondary brain injury (edema, increased intracranial pressure, ischemia, brain stem hemorrhages),—documented and assessed both radiologically and by neuropathological examination—agreed “chronologically” with aneurysm rupture occurring during the attack. The studies by Krauland [4,12,28] and other collaborators remain an important guide for diagnostic choices and should be more easily accessible, and also in English. Cases such as this are rare and should be investigated further to clarify the mechanism underlying the development and rupture of the posterior communicating artery. In addition, Leestma, who studied 67 similar case reports, many of them from Krauland—distinguished in the posterior communicating artery a site of unspecific injury and one susceptible to total avulsion [29]. These findings could guide the pathologist towards the traumatic or non-traumatic nature of the aneurysm. Several authors agree that the shape and location can suggest the nature of the aneurysm [30,31]. If the proportions of small and normal aneurysm components are considered, the risk of rupture can be estimated. Therefore, morphological alterations could also be used to support a persistent aneurysm [30,31]. For example, a small neck that reduces blood flow within the aneurysm could be an indicator of endothelial dysfunction [30,32]. However, the most important factor to assess a pre-existing aneurysm remains size [32,33]. Conversely, in TICAs, it should be assessed if central or peripheral vessel rupture corresponds to nearby rigid structures [34,35].

Several studies have focused on the study of aneurysm walls. Recently, inflammatory and apoptotic mechanisms involved in the process of non-traumatic aneurysm were investigated [36]. Several microRNAs and signaling proteins have been observed to regulate the apoptotic pathway. Promising markers of pre-existing intracranial aneurysms could be the overexpression of miR-29, cypD mRNA, caspases -3, -8, -9, cytochrome c, and myeloid cell leukaemia 1 (Mcl-1) in smooth muscle cells (VSMCs) [37,38].

According to Fan, W. et al., aneurysm size could correlate with increased CD14 and TNF-alpha [39]. Studies conducted for clinical and therapeutic purposes in the forensic field have often become a forensic tool for highlighting disease. Although many studies are still experimental, several microRNAs look very promising [40] to be useful not only for highlighting lesion viability but also for understanding the cause of death [41]. However, more evidence is need for their application in forensic field. Further studies in humans should be conducted to understanding the mechanisms underlying aneurysm rupture.

## 4. Conclusions

Our case confirms that TICAs of the posterior communicating artery can be the direct consequence of the violent rotation of the head and neck with distension of the intracranial vessels. When the victim abuses alcohol, drugs, or other substances known to facilitate the formation and rupture of an aneurysm, a detailed assessment should be made (e.g., circumstances, medical records, injuries, CT angiography, and DSA). Moreover, in fatal cases, the histological features of the rupture site allow us to date the rupture of the TICA. The importance of this study is underscored by the position of the TICA, which can lead to multiple forensic interpretations with obvious judicial consequences. The rupture of a pre-existing aneurysm after an assault can influence a judicial trial decision of homicide without homicidal intention. With regard to that, this article could be useful for the differential diagnosis of traumatic and non-traumatic aneurysms.

## Figures and Tables

**Figure 1 healthcare-12-00192-f001:**
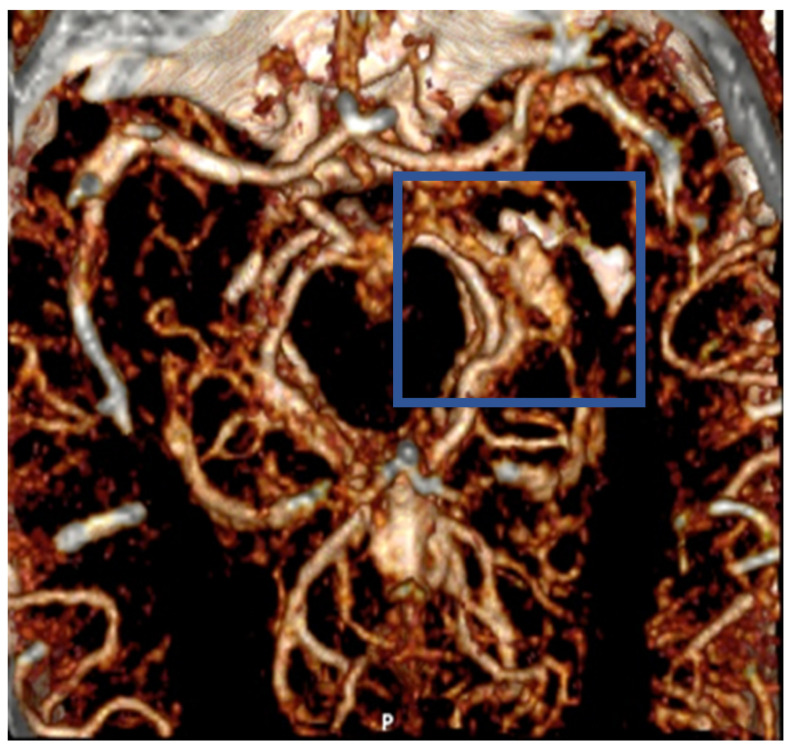
CT-angiography performed in the emergency room showed 5 mm TICA (blue square).

**Figure 2 healthcare-12-00192-f002:**
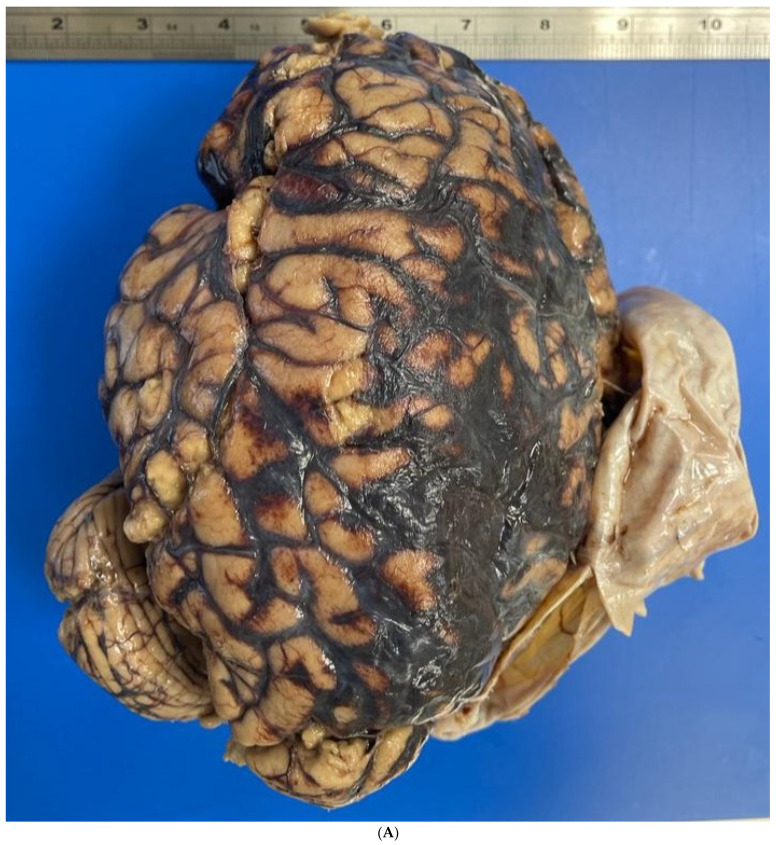
(**A**) Subarachnoid hemorrhage, small rounded subpial hemorrhage, swollen brain with flattened gyri and narrowed sulci. (**B**) Hemorrhages of the corpus callosum and thalamic nuclei. (**C**) Circle of Willis with a large blood clot involving the left internal carotid artery and the posterior communicating artery and part of metal coil. (**D**) Reddish-black hemorrhage in the midbrain; secondary injury due to supratentorial pressure (arrow).

**Figure 3 healthcare-12-00192-f003:**
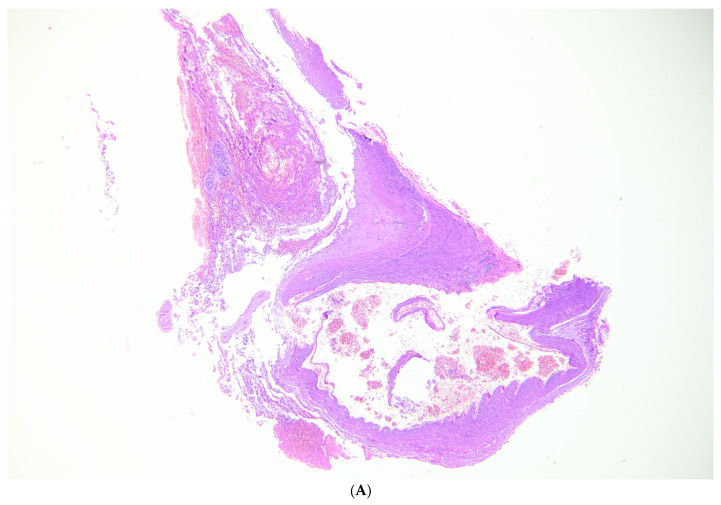
(**A**) Microscopy in TICA of left posterior communicating artery. (**B**) Rupture of the aneurysm with neutrophils, fresh red blood cells, fibrin, and no signs of hemosiderin (Masson’s trichrome on the left and H&E on the right (**C**)). (**D on the left**): Parietal cortex with necrosis, hemorrhage, and edema with some neutrophils inside (H&E); same characteristics in the basal nuclei (**E on the right**).

## Data Availability

The data that support the findings of this study are available from the corresponding author upon reasonable request.

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
