# Peer review of "Traumatic Aneurysm Involving the Posterior Communicating Artery"

_healthcare, 2024, doi:10.3390/healthcare12020192_

Round 1

Reviewer 1 Report

Comments and Suggestions for Authors

The authors present a comprehensive case study in the realm of forensic neuropathology. The case is well-documented, providing extensive details about the patient's condition, medical interventions, and autopsy findings. This comprehensive approach enriches the understanding of traumatic intracranial aneurysms (TICA).

The article aims to propose a diagnostic algorithm for managing suspected traumatic aneurysms in forensic settings, which is a significant contribution to the field. The study also effectively integrates forensic science, neuropathology, and radiology, demonstrating a multidisciplinary approach to solving complex medical-legal cases.

However, the manuscript, while detailed, is based on a single case, so the high soundness within the text should be quite reduced.

In addition, the article could benefit from a comparative analysis with other TICA cases or similar neurological conditions. Discussing the broader implications of the findings in relation to general forensic neuropathology practices could enhance the article's impact from a mere case report.

Comments on the Quality of English Language

The minor suggetion is to break down complex sentences into shorter, in order to improve readability.

Author Response

Dear reviewer,

i really appreciatd your comments.

 The manuscript has been modified and similar cases of TICA were included.

Additionally, medical English has been revised.

You will find changes in yellow and deleted words in red.

Although this is a single case report, I hope it can be valorised in this form.

All the best and Happy New Year, 

Gabriele

Reviewer 2 Report

Comments and Suggestions for Authors

This is an interesting article regarding a potentially traumatic aneurysm on the posterior communicant artery.

The abstract should present all relevant information, which is not the case. In this case, there was a metal coil, which is not mentioned in the abstract, and whose presence, integrity and anatomical relations should have been described in the case report.

The introduction provides sufficient background. The materials and methods are too long for a case presentation and describe standard  a autopsy protocol, which seems superfluous. 

Figure 3A and B should be removed, as it does not bring enough new data compared to the text from the manuscript.

Comments on the Quality of English Language

Some corrections needed

Author Response

Dear reviewer,

I really appreciate your comments.

In the uploaded file you will find  the description of the coil.

Futhermore, the materials and methods have been reduced.

Figure 3A and B have been removed. Also Medical english has been revised.

You will find changes in yellow and deleted words in red.

Our hope is that this new form can be an appreciable scientific contribution.

I wish you happy holidays and good work.

All the best,

Gabriele

Reviewer 3 Report

Comments and Suggestions for Authors

The authors describe a case of fatal haemorrhage from a traumatic aneurysm of the posterior communicating artery, and have added material beyond that expected from a simple case report, including a diagnostic algorithm to assist future workers faced with the same forensic problem. Some of the material is speculative: for instance, lines 204 to 230 are about potential utilisation of detailed molecular biological approaches, which might be dealt with by a single sentence and a few references. In contrast, there is a substantial literature (Huhtakangas et al; Shibata et al) about non-traumatic aneurysms of the posterior communicating arteries that should be consulted for context. There are also other case reports that deal with traumatic aneurysms that need to be consulted (Takahashi et al; Zhang et al), including a true aneurysm that ruptured during angiograph (Guo and Wu).

The authors place appropriate emphasis on the clinicopathological correlation, including correlation with radiology, and it would greatly enhance their paper to have one or two well-chosen radiographs, to add to their excellent illustrations..

Here is a list of potential minor improvements and other constructive criticisms

-        Line 2: given that there is more than just a case in the subsequent paper, I suggest that deleting the first 8 words would provide a more helpful title

-        Line 3 and elsewhere: substitute “communicating” for “communicant”

-        Line 40: substitute “dilation” for “vasodilation”; vasodilation is used for reversible dilation, for example as a response to autonomic signalling

-        Lines 40 -41 : substitute “focal wall weakness” for “partial wall rupture”; the latter implies some single traumatic event

-        Line 46: Strictly speaking, many dissecting aneurysms are not dilations, and I always use the more accurate term of “dissecting haematoma”

-        Line 48: Blood from the false lumen can rupture internally, where there is no prospect of dilation, or externally, with the potential development of a false aneurysm

-        Line 62: It is not usual to have Materials and Methods in a Case Report, and this section could usefully be incorporated into Section 3, Case Report

-        Line 63: substitute “detailed examination” for “deep learning”

-        Line 64: substitute “radiological” for “instrumental”

-        Line 66: substitute “in” for “on”; references 9 and 10 refer to infant studies, and are not appropriate comparisons in an adult case

-        Line 67: substitute “tissue” for “tissues”; delete both hyphens

-        Line 68: substitute “galea aponeurotica” for “galea capitis”

-        Line 71: substitute “removal” for “remotion”

-        Lines 73 to 78: This appears to be a description of technique, rather than specific finding; I am surprised there was no blood staining within the lateral ventricles

-        Line 78: substitute “allows” for “make”, and “to eviscerate” for “eviscerate”

-        Line 79: It is important to know the duration of fixation, and how fixation was achieved. The conventional method is to suspend the brain in a large volume of fixative, using the basal vessels as a suspension point

-        Lines 80 to 84: This is history, and does not belong in this section

-        Line 89: substitute “Perl’s” for “Pearl’s”

-        Line 93: substitute “were” for “was”

-        Line 99: substitute “highlighted” for “highlight”

-        Line 102: There should be brief comment on the concentrations of cocaine and alcohol, indicating their usual effect in human subjects

-        Line 109: substitute “contrecoup” for “contreculp”

-        Line 116, and elsewhere: substitute “circle of Willis” for “Polygon of Willis”. I agree that polygon seems more sensible, but we all use circle

-        Lines 122,141: The term “brain shift” is generally used for the bulk movement of the brain across the tentorial space

-        Line 135: Not sure whether I can see notching due to compression at the tentorium, but if it exists, highlighting with an arrow would help

-        Line 163 and elsewhere: “Phenomenal” is used (perhaps illogically) in English to indicate an extreme, so a “phenomenal haemorrhage” would be a very large volume of blood; “chronological” may be better

-        Line 182: substitute “trichrome” for “Trichromic”

-        Line 197: A reference, or references, to the work of Krauland should be given

Guo S, Wu X. An unruptured posterior communicating artery aneurysm ruptured during angiography: A case report. Medicine (Baltimore). 2019 Nov;98(44):e17785.

Huhtakangas J, Lehecka M, Lehto H, Jahromi BR, Niemelä M, Kivisaari R. CTA analysis and assessment of morphological factors related to rupture in 413 posterior communicating artery aneurysms. Acta Neurochir (Wien). 2017 Sep;159(9):1643-1652.

Shibata A, Kamide T, Ikeda S, Yoshikawa S, Tsukagoshi E, Yonezawa A, Takeda R, Kikkawa Y, Kohyama S, Kurita H. Clinical and Morphological Characteristics of Ruptured Small (<5 mm) Posterior Communicating Artery Aneurysms. Asian J Neurosurg. 2021 May 28;16(2):335-339. doi: 10.4103/ajns.AJNS_495_20.

Takahashi A, Kamiyama H, Imamura H, Kitagawa M, Abe H. "True" posterior communicating artery aneurysm--report of two cases. Neurol Med Chir (Tokyo). 1992 Jun;32(6):338-41.

Zhang C, Chen H, Bai R. Traumatic aneurysm on the posterior cerebral artery following blunt trauma in a 14-year-old girl: case report. Neuropediatrics. 2011 Oct;42(5):204-6.

Comments on the Quality of English Language

The use of language is generally grammatical, but there are many words that are used idiosyncratically, and not in the manner usually employed in medical English. I assume that this is a product of digital translation, and I have suggested changes, alongside other constructive criticisms, to more conventional words: English is not my first language, but I have worked in England for 52 years, which allows a degree of insight

Author Response

Dear reviewer,

I really appreciated the sincerity of your comments.

In the uploaded file you will find substantial changes to lines 228-272 (previusly 204-230). Futher cases were taken into consideration including those indicated by you.

Your suggestions on form and medical terminology were really precious. You will see that i followed your suggegestion point by point.

You will find changes in yellow and deleted words in red. 

I sincerly thanks you for yuor great contribution.

 I hope we will sign this manuscript together.

I wish you happy holidays and good work.

All the best,

Gabriele.

Round 2

Reviewer 3 Report

Comments and Suggestions for Authors

The authors have worked hard to improve this manuscript, and are to be congratulated on the many improvements. However, as often happens, widespread corrections lead to new problems, or make pre-existing ones more visible. I have a shorter (fortunately) list of new constructive suggestions.

 -             Line 20: substitute “died” for “dies”

 -             Line 31: substitute “assessment of” for “manage”

 -             Line 71: substitute “as well as the” for “as the”

-              Line 108 to 109: Cocaine is unstable in post mortem body fluids, and interpretation needs to be done with care; it is not possible to be completely sure that the concentration wasn’t greater at the time of death than that found on chemical assay; most toxicological reports include comments on metabolites such as benzoylecgonine, which can give some assistance with interpretation (though conclusive interpretations are rarely possible). The authors may want to add a short comment about this in line 187

-              Line 109: Drunkenness is never justifiable! Try “This alcohol concentration would be consistent with…

 -             Line 117: substitute “coup” for “cuop”

As an aside, the authors might find passing interest in the fact that this reviewer has worked for decades in the Department that published the first case where intracerebral haemorrhage was associated with cocaine misuse.

Comments on the Quality of English Language

There are also frequent grammatical idiosyncrasies, but they can be dealt with by  a subeditor.

Author Response

Dear reviewer,

Thank you again. I made the new corrections in green.

English has also been revised.

Even though I don't know your identity, your long experience in this field can be understood.

All the best.

Gabriele